# Antifungal Activity of Chitosan against *Histoplasma capsulatum* in Planktonic and Biofilm Forms: A Therapeutic Strategy in the Future?

**DOI:** 10.3390/jof9121201

**Published:** 2023-12-15

**Authors:** Raimunda Sâmia Nogueira Brilhante, Anderson da Cunha Costa, Jacó Ricarte Lima de Mesquita, Gessica dos Santos Araújo, Rosemeyre Souza Freire, João Victor Serra Nunes, Augusto Feynman Dias Nobre, Mirele Rodrigues Fernandes, Marcos Fábio Gadelha Rocha, Waldemiro de Aquino Pereira Neto, Thomas Crouzier, Ulrike Schimpf, Rodrigo Silveira Viera

**Affiliations:** 1Department of Pathology and Legal Medicine, School of Medicine, Specialized Medical Mycology Center, Postgraduate Program in Medical Sciences, Federal University of Ceará, Rua Barão de Canindé, 210, Montese, Fortaleza 60425-540, CE, Brazil; andersoncosta.biomed@gmail.com (A.d.C.C.); feynmandias@gmail.com (A.F.D.N.); mirele.mrf@gmail.com (M.R.F.); wapn@det.ufc.br (W.d.A.P.N.); 2São José Hospital, Fortaleza 60455-610, CE, Brazil; jacomesquita@hotmail.com; 3Postgraduate in Veterinary Sciences, Faculty of Veterinary, State University of Ceará, Dr. Silas Munguba Avenue, 1700, Itaperi Campus, Fortaleza 60714-903, CE, Brazil; gearaujo3@gmail.com (G.d.S.A.); mfgrocha@gmail.com (M.F.G.R.); 4Analytical Center, Department of Physics, Federal University of Ceará, Fortaleza 60020-181, CE, Brazil; rosefreire@centralanalitica.ufc.br (R.S.F.); jvictorsn@centralanalitica.ufc.br (J.V.S.N.); 5KTH Royal Institute of Technology, School of Engineering Sciences in Chemistry, Biotechnology and Health, Department of Chemistry, Division of Glycoscience, AlbaNova University Center, 106 91 Stockholm, Sweden; thocro@dtu.dk (T.C.); ulrikeschimpf@gmail.com (U.S.); 6Department of Chemical Engineering, Federal University of Ceará, Fortaleza 60440-900, CE, Brazil; rodrigo@gpsa.ufc.br

**Keywords:** nanoparticles, natural antimicrobials, antifungals, histoplasmosis

## Abstract

Histoplasmosis is a respiratory disease caused by *Histoplasma capsulatum*, a dimorphic fungus, with high mortality and morbidity rates, especially in immunocompromised patients. Considering the small existing therapeutic arsenal, new treatment approaches are still required. Chitosan, a linear polysaccharide obtained from partial chitin deacetylation, has anti-inflammatory, antimicrobial, biocompatibility, biodegradability, and non-toxicity properties. Chitosan with different deacetylation degrees and molecular weights has been explored as a potential agent against fungal pathogens. In this study, the chitosan antifungal activity against *H. capsulatum* was evaluated using the broth microdilution assay, obtaining minimum inhibitory concentrations (MIC) ranging from 32 to 128 µg/mL in the filamentous phase and 8 to 64 µg/mL in the yeast phase. Chitosan combined with classical antifungal drugs showed a synergic effect, reducing chitosan’s MICs by 32 times, demonstrating that there were no antagonistic interactions relating to any of the strains tested. A synergism between chitosan and amphotericin B or itraconazole was detected in the yeast-like form for all strains tested. For *H. capsulatum* biofilms, chitosan reduced biomass and metabolic activity by about 40% at 512 µg/mL. In conclusion, studying chitosan as a therapeutic strategy against *Histoplasma capsulatum* is promising, mainly considering its numerous possible applications, including its combination with other compounds.

## 1. Introduction

*Histoplasma capsulatum* is a dimorphic fungus whose saprophytic filamentous form is found in soils, especially those containing bird and bat feces, and whose parasitic yeast form is found in humans and animals [1,2]. Inhalation of conidia or mycelial fragments of *H. capsulatum* in contaminated environments is the primary mode of infection of both humans and animals, with these fungal elements readily carried through the air. Furthermore, on rare occasions, direct transmission of the parasitic form can occur through tissue transplantation or laboratory accidents [3,4]. Currently, histoplasmosis is one of the most prevalent systemic mycoses in the Americas [5]. In recent decades, systemic and invasive fungal infections have become more common, especially among immunocompromised patients. Annually, there are an estimated 300 million cases of severe fungal infection and 1.6 million deaths worldwide [6,7]. This situation primarily arises from the rising number of immunocompromised patients, such as those with AIDS, cancer, and transplant recipients, in addition to the increase in drug resistance, and the emergence and re-emergence of pathogens like *Histoplasma capsulatum* [7,8].

*H. capsulatum* displays a variety of virulence attributes that affect its interaction with host immune cells, allowing it to evade the immune response and replicate in a new environment, including features such as thermal dimorphism and biofilm formation [9,10]. The in vitro biofilm-forming ability of this fungal species has been reported for the yeast form [10,11] and has also been reported for other dimorphic fungi such as *Sporothrix* spp. and *Paracoccidaides* spp. [12,13].

Biofilms are microbial communities that firmly adhere to biotic or abiotic surfaces and are protected by an extracellular polymeric matrix composed of polysaccharides, proteins, nucleic acids, and lipids. These cells present more phenotypic, genetic, and structural heterogeneity in comparison with planktonic cells [14,15]. In the host, biofilms play a critical role in the development of the infection, making their formation an important virulence factor, since it confers greater resistance to specific mediators of immune response and increased resistance to antimicrobial agents. In the environment, this form of growth prevails over planktonic growth and provides protection for the fungal structures against environmental aggressors, such as Ultraviolet (UV) radiation, dehydration, extreme temperatures, and chemical agents [14,16]. The in vitro studies conducted with the goal of characterizing these fungal biofilms have employed various techniques, including the use of XTT salt (2,3-bis(2-methoxy-4-nitro-5-sulfophenyl)-5-[carbamoylphenylamino]-2-tetrazolium hydroxide) to assess metabolic activity during biofilm development and antifungal sensitivity testing. Additionally, crystal violet staining is utilized to quantify biomass [17,18]. Biofilm evaluations can also be conducted using specialized microscopic techniques, including scanning electron microscopy, to examine fungal biofilms’ morphology [19].

At present, the first-line histoplasmosis treatment is the administration of itraconazole and amphotericin B. However, relapses and refractory infections have been reported, predominantly due to biofilm formation [20,21]. In this context, developing target drugs is challenging [22].

Biopolymers like chitosan, a linear polysaccharide originating from partially deacetylated chitin, have been employed to counteract both planktonic and biofilm cells of diverse microorganisms, including other fungi such as *Candida* spp. [23], Aspergillus fumigatus. [24], and *Sporothrix brasiliensis* [23]. This biopolymer has gained extensive usage as an antimicrobial agent due to its biocompatibility, biodegradability, low toxicity, and significant functional potential, attributable to its amino and hydroxyl groups [25]. 

In a previous study conducted by our research group, chitosan with three different molecular weights (high, medium, and low) was investigated against the planktonic and biofilm forms of *Sporothrix brasiliensis* [23]. The results demonstrated that low molecular weight chitosan (LWC) had the most potent inhibitory effect against both the planktonic and biofilm forms of the fungus since smaller quantities of chitosan were needed to inhibit the growth of *S. brasiliensis*. In the present study, we evaluated the antifungal activity of low molecular weight chitosan, its interactions with antifungal drugs against planktonic cells, and its effect on biofilm formation in both filamentous and yeast forms of *Histoplasma capsulatum*.

## 2. Materials and Methods

### 2.1. Low Molecular Weight Chitosan Samples

The low molecular weight (LMW 448869) chitosan used in this study was obtained from Sigma-Aldrich (Saint Louis, MO, USA). The LWC was characterized regarding its molecular weight (MW) and deacetylation degree (DD) by viscometry and potentiometric degree, respectively, in a previous study described by us (Garcia et al., 2020 [23]). The MW was 206.4 kg.mol^−1^ and the DD was 79%. 

### 2.2. Fungal Culture

A total of 20 isolates of *H. capsulatum* were included in this study. All isolates were obtained from the culture collection of the Specialized Center for Medical Mycology (CEMM) of Federal University of Ceará, Brazil. The isolates were stored in physiological saline at 4 °C or on Potato Dextrose Agar supplemented with 10% DMSO (Dimethyl Sulfoxide, Sigma, St. Louis, MO, USA) at −20 °C. Before storage, these isolates underwent identification through mycological, immunological, and molecular techniques. For the assessment of purity and viability of the isolates taken from the stock, micromorphological analysis was conducted after inoculation on Potato Agar and incubation for 7–15 days at room temperature (−28 °C). The cultivation of *H. capsulatum* in the yeast phase was achieved from cultures in the filamentous phase subjected to successive subcultures on Sabouraud Agar and Brain Heart Infusion (BHI) Agar supplemented with 10% sheep blood, followed by incubation at 35 °C. The procedures were performed in a class II biological safety cabinet in a biosafety level 3 laboratory. 

### 2.3. Preparation of Chitosan and Control Drugs

Chitosan (1 g) was dissolved in 100 mL of 1% (*v*/*v*) glacial acetic acid solution (Panreac, Barcelona, Spain) and stored under refrigeration. Amphotericin B (AMB) and itraconazole (ITC) (Sigma, St. Louis, MO, USA) were used as control drugs and were dissolved in dimethyl sulfoxide (DMSO) (Sigma-Aldrich), according to the Clinical and Laboratory Standards Institute [26]. Subsequently, AMB and ITC were prepared in Roswell Park Memorial Institute (RPMI) 1640 medium (Sigma, St. Louis, MO, USA) buffered to pH 7.0 with 0.165 M MOPS (3-[N-Morpholino] propane sulfonic acid) (Sigma-Aldrich).

### 2.4. Antifungal Susceptibility Test in Planktonic Cells of H. capsulatum

The sensitivity tests were conducted with the 20 strains of *H. capsulatum* using the broth microdilution technique, based on the reference methods of document M38-3rd edition [26] for the filamentous form and M60-2nd edition [27]. for the yeast form, with adaptations. The compounds were tested at final concentrations of 1–1024 µg/mL for chitosan and 0.03–16 µg/mL for amphotericin B and itraconazole. The inoculants were obtained from growth of *H. capsulatum* cultures on potato dextrose agar (PDA). The cells were suspended in sterile saline, after each fungal suspension was adjusted to 1 on the McFarland for the filamentous form and 0.5 on the McFarland scale for the yeast form. The 1:10 dilutions were made in RPMI 1640 buffered to pH 7.0 with MOPS 0.165 M to obtain inoculants with a final concentration from 1 × 10^5^ to 5 × 10^5^ CFU/mL for the filamentous form and 1 × 10^3^ to 5 × 10^3^ CFU/mL for the yeast form. The tests were performed in duplicate, and the results were read visually after the incubation period of 72 h at 35 °C for the filamentous form and 96 h at 37 °C for the yeast form. [11] We considered the MIC of chitosan, itraconazole, and amphotericin B as the lowest concentration capable of inhibiting the visual growth of fungi compared to growth control by 50, 80, and 100%, respectively. *Candida parapsilosis* ATCC 22019 and *Candida krusei* ATCC 6258 were used as quality controls for the experiments, which were carried out in duplicate at two different time points.

### 2.5. Susceptibility Test of Biofilm Formation

Flat-bottomed 96-well plates were employed for the formation of filamentous and yeast forms of biofilms. A 200 µL aliquot of fungal suspension at 1 × 10^6^ conidia/mL (filamentous form) or 2 × 10^5^ cells/mL (yeast form) prepared in RPMI 1640 medium was added to 96-well microplates and incubated in an oven at 35 °C (filamentous form) or 37 °C (yeast-form) for 24 h to allow for cellular adhesion. Chitosan, amphotericin B, and itraconazole were tested against planktonic strains at concentrations of MIC, 10 × MIC, and 50 × MIC. After the 24-h incubation period for microorganisms to adhere to the wells, the supernatant in each well was removed and the biofilms were washed with sterile PBS once. The wells then received 200 µL of RPMI medium (controls) or drugs at the concentrations mentioned earlier. The plates were incubated for 72 h at 35 °C (filamentous form) or 37 °C (yeast form). The susceptibility of sessile cells to chitosan and antifungals was analyzed by quantifying biomass and metabolic activity using violet crystal staining and the MTT reduction assay along with morphological visualization using scanning electron microscopy, respectively [28]. 

### 2.6. Biomass Quantification Assay

The biomass of the biofilms was quantified using the crystal violet staining technique. The wells containing biofilms were washed twice with PBS and fixed with 100% methanol for 5 min (GQ—Grupo Química, São Paulo, Brazil). Then, the methanol was removed, and the wells were air-dried at room temperature. Subsequently, 200 μL of a 0.3% crystal violet solution was added to each well for 20 min. After this period, the wells were washed twice with distilled water. The wells were then destained by adding a 33% acetic acid solution. The remaining content in each well was transferred to the well of the new plate, and the absorbance was measured at 540 nm using a spectrophotometer [17]. All tests were performed in duplicate at two different time points.

### 2.7. Metabolic Activity Quantification Assay

The metabolic activity of the biofilms was quantified using the 2,3-bis-(2-methoxy-4-nitro-5-sulfophenyl)-5-[(phenylamino) carbonyl]-2H-tetrazolium (MTT; Sigma, Darmstadt, Germany) reduction assay [17]. This assay was performed using a stock solution of MTT (1 µg/mL in PBS) and menadione (1 mM in ethanol). To all wells containing biofilm, 125 μL of MTT solution was added, and after 3 h of incubation at 35 °C, protected from light, the color change was measured with a spectrophotometer at 492 nm. The minimum inhibitory concentration in biofilm (MICB) was determined as the lowest concentration capable of inhibiting 50% (MICB50) or 80% (MICB80) of the biofilm’s metabolic activity in comparison with the drug-free control’s activity [29]. All tests were conducted in duplicate at two different time points.

### 2.8. Evaluation of the Morphology and Structure of H. capsulatum Biofilms

The LWC effect on the morphology and structure of mature *H. capsulatum* biofilms was investigated using scanning electron microscopy (SEM). Mature biofilms were formed as described in the previous section in 24-well plates and treated with MICB80 of chitosan, as previously described.

For SEM analyses, the biofilms were fixed with 500 μL of 2.5% glutaraldehyde solution (Sigma-Aldrich, St Louis, MO, USA) in cacodylate buffer (0.15 M) (Electron Microscopy Sciences, Hatfield, PA, USA) and 0.1% Alcian blue (Sigma-Aldrich, São Paulo, Brazil), at 4 °C, overnight, to preserve fungal structures. Subsequently, biofilms were washed with cacodylate buffer twice, followed by dehydration in baths containing ascending ethanol concentrations (50, 70, 80, 95, and 100%) for 10 min at each dehydration level, repeating the dehydration with 100% ethanol once more. After drying, biofilms were dehydrated with hexamethyldisilazane (DMSO) (Sigma, St. Louis, MO, USA) for 30 min and stored at 35 °C overnight. Then, coverslips were coated with 10 nm of gold (Emitech Q150T, Shanghai, China) and observed with a Quanta FEG 450 scanning electron microscope (Thermo fisher, São Paulo, Brazil) under a high vacuum mode at 20 kV using a secondary electron detector (Thermo fisher, São Paulo, Brazil) [17].

### 2.9. Pharmacological Interaction—Checkerboard

To evaluate interactions between chitosan and antifungals (AMB, ITC), strains of *H. capsulatum* were selected randomly in both filamentous and yeast forms, and were submitted to the checkerboard broth microdilution method with a 7-row by 11-column configuration [30]. The checkerboard assay was performed as described by Brilhante et al. (2020) [17], with adaptations. The combinations of chitosan (8 to 512 µg/mL) with amphotericin or itraconazole (0.01 to 16 µg/mL) were evaluated. In the assays with planktonic cells, combinations of variable concentrations of chitosan ranging from 4 to 256 µg/mL for the filamentous form and 0.5 to 32 µg/mL for the yeast form of *H. capsulatum* were used, combined with 0.007 to 8 µg/mL of AMB or ITC. Inoculums were prepared to a final concentration of 0.4–5 × 10^4^ cfu/mL (filamentous form) or 0.5–2.5 × 10^3^ cfu/mL (yeast form) in RPMI 1640 medium as previously described. The microplates were incubated at 35 °C (filamentous form) or at 37 °C for 96 h (yeast form) or 72 h, and the MICs were defined following the antifungal reading points defined in the CLSI [11]. MIC values were defined as the lowest concentration capable of inhibiting 100% (amphotericin) or 80% (itraconazole) of the growth of fungi by the combination between chitosan and amphotericin B or itraconazole. The MIC values obtained were used to determine the fractional inhibitory concentration index (FICI). The interactions were defined as synergistic when FICI ≤ 0.5, indifferent when 0.5 < FICI ≤ 4, and antagonistic when FICI > 4 [31]. All tests were conducted in duplicate at two different time points.

### 2.10. Statistical Analyses

Experimental results are expressed as means ± standard deviations (SD). Student’s *t*-test and one-way analysis of variance (ANOVA) were applied for comparisons where the data exhibited asymmetry, while the nonparametric Wilcoxon or Friedman test was used otherwise, followed by Dunn’s post-test. Differences were considered statistically significant at *p* < 0.05. The statistical analyses were conducted using GraphPad Prism 7.0 (GraphPad Software, San Diego, CA, USA).

## 3. Results

### 3.1. Antifungal Susceptibility

Chitosan showed inhibitory activity against the *H. capsulatum* in its filamentous and yeast forms (Table 1 and Table 2, respectively). The MIC values of chitosan in the filamentous form of *H. capsulatum* ranged from 32 to 128 µg/mL. For the antifungal drugs, the MIC values obtained ranged from 0.25 to 2 µg/mL for amphotericin B and from 0.25 to 1 µg/mL for itraconazole (Table 1).

The MIC values of chitosan for the yeast form of *H. capsulatum* ranged from 8 to 64 µg/mL. The MIC values observed for antifungal drugs ranged from 0.03 to 0.5 µg/mL for amphotericin B and 0.25 to 0.5 µg/mL for itraconazole, as shown in Table 2.

### 3.2. Pharmacological Interactions

The MIC values of the drugs alone and combined with chitosan and the FICI values obtained for the filamentous and yeast forms of *H. capsulatum* are reported in Table 3 and Table 4, respectively. For the combinations of chitosan with antifungals against the filamentous form of *H. capsulatum*, synergy was observed with ITC involving two of the tested strains. In these combinations, the MICs of the antifungals were 4 to 16 times lower than those observed individually. On the other hand, the combinations with AMB were all indifferent, and no antagonistic interaction was observed in the combination with either drug. 

Pharmacological interactions between LWC and the antifungals AMB and ITC were also tested. MIC results for drugs, individually and in combination, against *H. capsulatum* in its filamentous form are presented in Table 3. LWC did not exhibit any antagonism and displayed synergy against some strains. LWC combined with AMB and ITC reduced the MIC by up to 32 times in some situations. The decreases in MIC are proportional to the MICs of the isolated drugs, with more modest reductions in the MICs of antifungals compared to the decreases observed in chitosan. Thus, demonstrating the advantageous combined effects.

The MIC values for antifungal drugs, either alone or combined with LWC, against yeast forms of *H. capsulatum* are presented in Table 4. Synergistic interaction between LWC and AMB or ITC was observed, resulting in a reduction of up to 16-fold in LWC MIC (Table 4). Similar to the filamentous form, no antagonism was observed, and all combinations exhibited synergistic results against all tested strains.

### 3.3. Evaluation of Chitosan Effects on H. capsulatum Biofilm Formation

Chitosan caused significant reductions in the biomass and metabolic activity of *H. capsulatum* biofilms in the filamentous form (Figure 1A). Biomass reductions were observed when exposing *H. capsulatum* biofilms to chitosan, demonstrating a dose-dependent effect [128 µg/mL causing 47% reduction] (*p* < 0.05). In the assessment of metabolic activity, a significant reduction of 30% was obtained from 32 to 512 µg/mL of chitosan (*p* < 0.05) (Figure 1B). There were no significant reductions among the different strains tested.

Chitosan also caused significant reductions in the biomass and metabolic activity of *H. capsulatum* biofilms in the yeast form (Figure 2). Reductions in biomass occurred when *H. capsulatum* biofilms were exposed to chitosan, with a dose-dependent relationship [32 µg/mL, resulting in 40% reduction] (*p* < 0.05). In the evaluation of metabolic activity, a notable reduction of 40% was achieved with a chitosan concentration range of 32 to 512 µg/mL (*p* < 0.05) (Figure 2B). No significant reductions were observed among the various tested strains.

### 3.4. Structural Analysis of Biofilms

SEM was performed to depict structural differences between the biofilms of *H. capsulatum* treated, or not, with chitosan (Figure 3A–C). In the absence of LMW chitosan, biofilms of *H. capsulatum* showed fungal cells that were organized into dense structures and composed of multilayers of associated cells besides an extracellular matrix (Figure 3A). Biofilms treated with chitosan at 4 × MIC100% and 8 × MIC100% exhibited less dense structures with less cells and less extracellular matrix (Figure 3B,C). After exposure to chitosan (128 µg/mL), a decrease in cell mass was observed (Figure 3B,C) compared to the control without chitosan exposure (Figure 3A). The cluster structure of blastoconidia was no longer evident after treatment (Figure 3B,C), and the cells that remained on the Thermanox slides displayed reduced viability, as can be observed in Figure 3B,C.

## 4. Discussion

Chitosan had an antifungal effect on the planktonic growth of *H. capsulatum*. It was able to inhibit the growth of the fungal pathogen in the filamentous and yeast forms. Previous studies have reported chitosan inhibition on the growth of other fungal pathogens, with results similar to those found in this study, such as *Aspergillus fumigatus* [32] and *Sporothrix brasiliensis* [23,33]. The results obtained for the antifungal drugs amphotericin B and itraconazole are in the same range of concentrations described by other studies [34,35,36].

Chitosan has attracted attention as a versatile biopolymer because of its cationic characteristics, biocompatibility, biodegradability, low toxicity, and good adsorption capabilities [37]. Numerous studies have examined the mechanism of chitosan [25,38,39]. It primarily functions through electrostatic interactions between its positively charged protonated amino groups and the negatively charged components of cell surfaces [40]. This interaction can be manifested through various pathways, and the broad molecular weight of chitosan (spanning from oligo-chitosan to low, medium, and high molecular weight) is one of the inherent factors that can affect its mechanism of action [41]. In a recent study, the antifungal properties of chitosan with different molecular weights were analyzed against three fungal species, *C. albicans*, *F. solani*, and *A. niger*. In that study, the authors found varying inhibition results depending on the fungal species and chitosan molecular weight, demonstrating that molecular weight can influence the observed antifungal activity [42]. According to the authors, low molecular weight chitosan may have a stronger impact on intracellular processes because it can more easily enter cells due to its small size and low zeta potential. Low molecular weight (LMW) chitosan can permeate the cell wall, interact with DNA, and impede mRNA synthesis and transcription. It has also been proposed that LMW chitosan can induce cell membrane rupture, representing a dual mechanism of action [23]. One hypothesis is that chitosan acts by permeabilizing the plasma membrane, leading to intracellular overproduction of reactive oxygen species (ROS), potentially inducing cell death [43].

In general, the yeast form of *H. capsulatum* exhibited MICs approximately four times lower than those obtained for the filamentous form in combination with the compounds tested in this study. As a dimorphic fungus, in nature, it exists in the filamentous (infectious) form, and when it enters the host, a temperature change triggers the fungal transition to the yeast form. A series of changes are necessary to become a yeast, from activating genes such (as the Ryp1 gene) to alterations in the fungal cell wall composition [44]. Therefore, this difference in susceptibility may be related to differences in the proportions of fungal cell wall constituents between filamentous and yeast forms, such as glucosamine and β-1,3-glucan [45]. The cell wall is essential for fungal growth since it provides stability and protection against osmotic stress. Furthermore, this structure is responsible for maintaining cellular shape. Changes in the cell wall structure have previously been linked to reduced susceptibility to amphotericin B in *Candida* spp. [46]. Thus, we hypothesize that differences in the cell wall composition between the filamentous and yeast forms of *H. capsulatum* may influence susceptibility to antifungal agents and chitosan.

The interaction effect of LWC and amphotericin B or itraconazole on the growth of *H. capsulatum* revealed that these combined drugs have synergistic effects in different combinations for *H. capsulatum* in the filamentous form, without any antagonistic effect. In these cases, the reduction in the MIC of the antifungal agent reached up to 16 times. In assays involving the yeast form, the results were more promising, and synergism between LWC and amphotericin B or itraconazole was detected for all strains tested. Synergism between chitosan and antifungal agents has previously been reported in experiments involving two *Candida* species [47].

Based on amphotericin B’s action in creating pores in fungal cell membranes and itraconazole’s effect of inhibiting ergosterol biosynthesis, leading to impaired fungal plasma membrane integrity, there may be an increased penetration of LWC into the cells [48]. The combined action on different biological processes or molecular targets could explain the synergism.

Chitosan caused reductions of approximately 50% in the rates of biomass and metabolic activity of biofilms. Micrographs obtained by SEM of *H. capsulatum* biofilms in the yeast form (the infective form) showed that untreated chitosan-free biofilms were robust, with higher cell numbers. In contrast, chitosan-treated biofilms had a decrease in the number of cells, demonstrating that chitosan acts on biofilms by killing cells. Other studies have demonstrated a reduction in biofilm formation in the presence of low molecular weight chitosan in *Candida albicans* [49] and *Sporothrix brasiliensis* [23], with reductions in biomass production and metabolic activity. Based on a clinical trial, another study investigated the effect of low molecular weight chitosan on *Candida albicans* biofilms in denture stomatitis. The authors concluded that chitosan has antifungal efficacy on biofilm formation in patients along with inherent biocompatibility, making it a promising candidate for use as an antifungal mouthwash [50].

It is known that during infection, the fungus *H. capsulatum* needs to maintain the stability of the cell membrane for its survival [51,52]. LMW chitosan is capable of causing cell membrane rupture, promoting an effect incompatible with the metabolic activity of the fungus [23]. It is also known that one of the identified effects of chitosan in general is the production of reactive oxygen species in cells. Previous studies have shown that oxidative stress is crucial in forming and maintaining biofilms [53,54]. Increased production of reactive oxygen species can lead to less biofilm with less extracellular matrix production [53]. Consequently, fungal pathogens, like *C. albicans*, have evolved various tactics to counteract the oxidative stress produced as a byproduct of aerobic respiration, thereby preserving redox homeostasis within cells [54,55]. In this sense, some pathogens use enzymatic mechanisms such as peroxidases to try to survive oxidative stress in the host [56,57]. Such effects may explain how polymers, whose action is the production of reactive oxygen species, cause reductions in the formation and maintenance of fungal biofilms, which could also explain the reductions found in the tests carried out in this study. 

## 5. Conclusions

Based on our in vitro results, the low molecular weight chitosan can inhibit strains of *H. capsulatum* in the filamentous and yeast forms, having pharmacological synergism with traditional antifungals. Moreover, chitosan inhibited the biomass and metabolic activity rates of *H. capsulatum* biofilms significantly, influencing the biofilm formation. In addition to our findings, low molecular wave chitosan has been extensively studied against various pathogens, both in vitro and in vivo, in recent years. Thus, the application of chitosan can be an innovative therapeutic strategy in the future.

## Figures and Tables

**Figure 1 jof-09-01201-f001:**
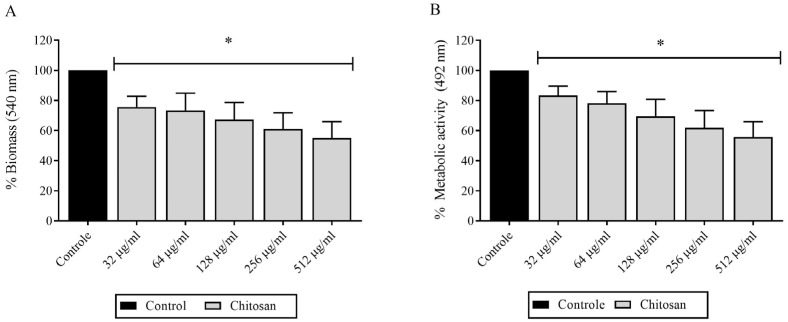
Metabolic activity and biomass of *H. capsulatum* biofilms in filamentous form. exposed to different concentrations of chitosan: (**A**) Biofilm biomass exposed to drug; (**B**) Metabolic activity of drug-exposed biofilm. * Significant difference compared to control (*p* < 0.05).

**Figure 2 jof-09-01201-f002:**
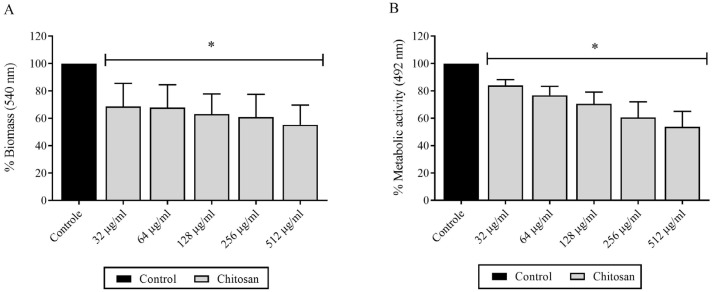
Metabolic activity and biomass of *H. capsulatum* biofilms in yest form exposed to different concentrations of chitosan: (**A**) Biofilm biomass exposed to drug; (**B**) Metabolic activity of drug-exposed biofilm. * Significant difference compared to control (*p* < 0.05).

**Figure 3 jof-09-01201-f003:**
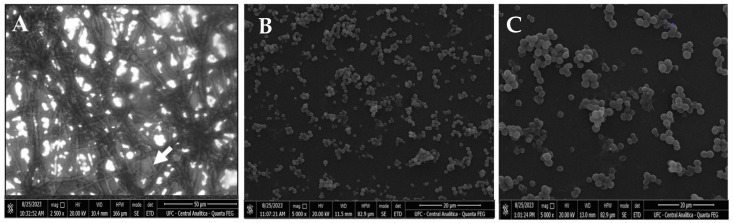
Structure and morphology of the *H. capsulatum* biofilm. SEM images of biofilms with untreated control, with an arrow pointing to the presence of extracellular polymeric substances (EPS): (**A**), and with treatment of 512 μg/mL chitosan (**B**) and 128 μg/mL chitosan (**C**). There is a reduction in the biofilm caused by the rupture of EPS in the biofilm, with only a few cells adhering to the surface after treatment with chitosan.

**Table 1 jof-09-01201-t001:** Minimum inhibitory concentration of amphotericin B, itraconazole, and chitosan against strains of *H. capsulatum* in the filamentous form.

Strains	Minimum Inhibitory Concentration (µg/mL)		
(CEMM Code)	AMB	ITC	CHI
CEMM 03-2-088	2	1	64
CEMM 03-2-090	0.5	0.25	32
CEMM 03-3-003	0.25	1	64
CEMM 03-3-026	2	1	64
CEMM 03-3-033	2	0.25	64
CEMM 03-3-034	0.5	0.25	32
CEMM 03-3-036	2	1	64
CEMM 03-3-037	2	1	64
CEMM 03-3-038	2	1	128
CEMM 03-3-039	0.5	0.03	64
CEMM 03-3-015	1	0.25	32
CEMM 03-3-072	2	1	64
CEMM 03-3-100	0.5	0.03	64
CEMM 03-3-024	0.5	0.03	128
CEMM 03-3-049	0.35	1	32
CEMM 03-3-066	2	0.25	32
CEMM 03-3-052	1	0.25	32
CEMM 03-3-070	0.5	1	64
CEMM 03-3-040	1	1	64
CEMM 03-3-053	0.5	0.03	32

CEMM: Center for Medical Mycology, AMB: amphotericin B, ITC: itraconazole, CHI: chitosan.

**Table 2 jof-09-01201-t002:** Minimum inhibitory concentration of amphotericin B, itraconazole, and chitosan against yeast form of *H. capsulatum*.

Strains	Minimum Inhibitory Concentration (µg/mL)		
(CEMM Code)	AMB	ITC	CHI
CEMM 03-2-088	0.25	0.5	8
CEMM 03-2-090	0.03	0.25	32
CEMM 03-3-003	0.5	0.5	64
CEMM 03-3-026	0.03	0.5	64
CEMM 03-3-037	0.03	0.5	8
CEMM 03-3-038	0.25	0.25	64
CEMM 03-3-039	0.5	0.5	32
CEMM 03-3-015	0.03	0.25	16
CEMM 03-3-072	0.5	0.5	16
CEMM 03-3-100	0.03	0.5	32
CEMM 03-3-033	0.25	0.25	16
CEMM 03-3-034	0.25	0.25	32
CEMM 03-3-036	0.25	0.25	32

CEMM: Center for Medical Mycology, AMB: amphotericin B, ITC: itraconazole, CHI: chitosan.

**Table 3 jof-09-01201-t003:** Minimum inhibitory concentration of chitosan in combination with antifungals against planktonic cells of *H. capsulatum*, in filamentous form, and fractional inhibitory concentration index.

CEMM Code	Isolated Drugs (µg/mL)			Combined Drugs (µg/mL)		FICI	
	CHI	AMB	ITC	CHI/AMB	CHI/ITC	CHI/AMB	CHI/ITC
CEMM 03-2-088	64	2	1	16/0.5	16/0.5	0,75 I	1.25 I
CEMM 03-2-090	128	0.5	0.25	64/2	32/1	1.5 I	0.75 I
CEMM 03-3-003	64	0.25	1	16/0.25	16/0.25	0.5 I	0.75 I
CEMM 03-3-026	64	2	1	32/0.0125	32/0.125	0.625 I	1.5 I
CEMM 03-3-033	64	2	4	32/1	16/0.25	1.5 I	0.75 I
CEMM 03-3-034	64	0.5	0.25	16/2	16/0.125	1.25 I	0.03 S
CEMM 03-3-036	64	2	1	16/2	16/0.25	1.25 I	0.03 S
CEMM 03-3-037	32	2	1	32/1	32/0.25	2.0 I	1.25 I

CEMM: Centre for Medical Mycology, AMB: Amphotericin B, ITC: Itraconazole, CHI: chitosan, FICI: fractional inhibitory concentration index, the interaction was classified as follows: synergistic (S; FICI ≤ 0.5), indifferent (I; 0.5 < FICI ≤ 4.0) or antagonistic (A; FICI > 4.0).

**Table 4 jof-09-01201-t004:** Minimum inhibitory concentration of chitosan in combination with antifungals against planktonic cells of *H. capsulatum* in the yeast form, and fractional inhibitory concentration index.

CEMM Code	Isolated Drugs (µg/mL)		Combined Drugs (µg/mL)		FICI	
	CHI	AMB	CHI/AMB	CHI/ITC	CHI/AMB	CHI/ITC
CEMM 03-2-088	32	0.25	4/0.03125	2/0.0078125	0.125 S	0.3125 S
CEMM 03-2-090	32	0.125	4/0.01625	2/0.0078125	0.125 S	0.3125 S
CEMM 03-3-003	32	0.5	2/0.03125	2/0.0078125	0.125 S	0.3125 S
CEMM 03-3-026	64	0.5	2/0.03125	2/0.0078125	0.125 S	0.3125 S
CEMM 03-3-033	64	0.5	2/0.03125	2/0.0078125	0.125 S	0.3125 S
CEMM 03-3-034	64	0.5	2/0.03125	2/0.0078125	0.125 S	0.3125 S
CEMM 03-3-036	32	0.5	2/0.03125	2/0.0078125	0.125 S	0.3125 S

CEMM: Center for Medical Mycology, AMB: Amphotericin B, ITC: Itraconazole, CHI: chitosan, FICI: fractional inhibitory concentration index, classified as synergistic (S; FICI ≤ 0.5), indifferent (I; 0.5 < FICI ≤ 4.0) or antagonistic (A; FICI > 4.0).

## Data Availability

Data are contained within the article.

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
