# Peer review of "Antifungal Activity of Chitosan against Histoplasma capsulatum in Planktonic and Biofilm Forms: A Therapeutic Strategy in the Future?"

_jof, 2023, doi:10.3390/jof9121201_

Round 1

Reviewer 1 Report

Comments and Suggestions for Authors The article is well written and designed. The biofilm is a current topic and manuscripts on this topic are welcome. However, I would have few suggestions in order to improve the quality of this article: 1. The Introduction section is too long, I would suggest to remove from this section the general information regarding the Histoplasma capsulatum. 2. In the Material and method section, please specify the producer of the Candida ATCC strains. 3. MIC should be expressed/calculated as µg/ml throughout the manucript.

Author Response

Comments and Suggestions for Authors

The article is well written and designed. The biofilm is a current topic and manuscripts on this topic are welcome. However, I would have few suggestions in order to improve the quality of this article: 1. The Introduction section is too long, I would suggest to remove from this section the general information regarding the Histoplasma capsulatum. 2. In the Material and method section, please specify the producer of the Candida ATCC strains. 3. MIC should be expressed/calculated as µg/ml throughout the manucript.

Author’s reply: Thank you very much for your considerations. As requested, we have revised the introduction to eliminate the excess of general information about Histoplasma capsulatum. In this new version, we are also including the producer of the ATCC strains used in our studies. We have also reviewed, and in the current version, the MIC is properly expressed in µg/ml throughout the manuscript. We believe that your contribution has significantly enhanced our work, and we appreciate the time devoted to correcting our manuscript.

Reviewer 2 Report

Comments and Suggestions for Authors

Dear Authors,

Thank you for submitting your manuscript, “Antifungal activity of chitosan against Histoplasma capsulatum in planktonic and biofilm forms: A therapeutic strategy in the future?”

These are my suggestions for corrections or requests for clarifications:

Line 31: “classical antifungal drugs showed a synergic effect, reducing chitosan’s MICs by 32 times” This is confusing “by 32 times” or is it better to mention the log reduction? Was this for the yeast-like form or hyphae?

Line 32: “demonstrating that these interactions did not exhibit any strains with antagonism” I suggest rewording.

Line 54-55: “high prevalence of the infection in that region” Which region?

Line 55-56: “At the moment histoplasmosis is one of the most prevalent systemic mycoses in the America” Did you mean was one of the most prevalent?

Line 56-61: “In recent decades, systemic and invasive fungal infections have become more common, especially among immunocompromised patients. Annually, there are an estimated 300 million cases of severe fungal infections and 1.6 million deaths worldwide [9,10]. This situation primarily arises from the rise in the count of immunocompromised patients, such as those with AIDS, cancer and transplant recipients, in addition to the increase in drug resistance, and the emergence and re-emergence of pathogens like Histoplasma capsulatum [10,11]. This section needs to be brought to the beginning of the introduction.

Line 71: "These cells" Which cells?

Line 76: "UV" please spell it out

Line 76 is repeat of line 64 with a variation: "H. capsulatum, in vitro biofilm-forming ability has been reported for filamentous and yeast forms [13,14]  "The in vitro biofilm-forming ability of this fungal species has been reported for the yeast form [13,14],

Line 79-83: "Within the host, biofilms play a crucial role in infection development, serving as a significant virulence factor by enhancing resistance against immune response mediators and antimicrobial agents. In the environment, this growth form predominates over planktonic growth,  safeguarding fungal structures from environmental threats, including UV radiation, dehydration, extreme temperatures, and chemical agents [21,22]" is again a repetition of same content from several line above it.

Line 134: Please explained how these 20 isolates were selected?

Line 151: Please explain with CLSI 2020 is used? and what do you mean by "with adaptation"?

Line 237: "CLSI" Which CLSI?

Line 159, 165, 192, 204, 242, "All tests were conducted in duplicate at two 242 different time points". Did you come up with exact same results? I could not find any statement or statistics of these duplicate in any section of the results. Please clarify. 

Table 1, 2, 3 and 4: "LMW" abbreviation is used for "LMW Chitosan" and it is not self explanatory and confusing to the reader. Please come up with another abbreviation to be more clear for "Chitosan" not the LMV form of it. 

Line 277, "In these combinations, the MICs of the antifungals were 4 to 16 times lower than those observed individually." This is confusing are you comparing it with the AMB and ITC alone or with LMV alone? It seems that AMB and ITC have very low MICs and in fact the combination barely improves the MIC compare to AMB or ITC alone. 

Table 3, line CEMM 03-0-34 under LMV/ITC it says "0.0S" Please clarify

Line 290: "LWC combined with AMB or ITC reduced the MIC by up to 32 times, demonstrating the combined effects." Same as it mentioned it seems that the AMB and ITC alone have very low MIC and LMV synergistic impact is very little. So comparison is perhaps was done with the LMV alone not with AMB and ITC alone. 

Line 300: ". Synergistic interaction between LWC and AMB or ITC was observed, resulting in up to a 16-fold reduction in antifungal MIC in the presence of LWC" Same issue as above. 

Line 397: "reduction in the MIC of the antifungal agent reached up to 16 times" Same issue as above.

Comments on the Quality of English Language

I suggest an overall English editing could improve the quality of the paper. 

Author Response

These are my suggestions for corrections or requests for clarifications:

Line 31: “classical antifungal drugs showed a synergic effect, reducing chitosan’s MICs by 32 times” This is confusing “by 32 times” or is it better to mention the log reduction? Was this for the yeast-like form or hyphae?

Author’s reply: We fully agree with your suggestion. To clarify the passage in this new version of the manuscript, we chose to present the result by providing its logarithmic reduction. Additionally, we have indicated that this reduction was observed in tests involving the fungus in its yeast form.

Line 32: “demonstrating that these interactions did not exhibit any strains with antagonism” I suggest rewording.

Author’s reply: As requested, we have rephrased the passage, as can be observed in the updated version of the manuscript.

Line 54-55: “high prevalence of the infection in that region” Which region?

Author’s reply: Here we are referring to the United States.

Line 55-56: “At the moment histoplasmosis is one of the most prevalent systemic mycoses in the America” Did you mean was one of the most prevalent?

Author’s reply: The most common endemic systemic mycoses are caused by thermally dimorphic fungi such as H. capsulatum, which currently ranks among the pathogens most frequently associated with systemic diseases. In the Americas, histoplasmosis remains among the group of systemic diseases with the highest prevalence to this day. In this revised version of the manuscript, we have clarified this passage.

Line 56-61: “In recent decades, systemic and invasive fungal infections have become more common, especially among immunocompromised patients. Annually, there are an estimated 300 million cases of severe fungal infections and 1.6 million deaths worldwide [9,10]. This situation primarily arises from the rise in the count of immunocompromised patients, such as those with AIDS, cancer and transplant recipients, in addition to the increase in drug resistance, and the emergence and re-emergence of pathogens like Histoplasma capsulatum [10,11]. This section needs to be brought to the beginning of the introduction.

Author’s reply: Thank you for the suggestion; we agree and have relocated the paragraph to the beginning of the section.

Line 71: "These cells" Which cells?

Author’s reply: Here we are referring to biofilm cells, we make it clearer in the new version.

Line 76: "UV" please spell it out

Author’s reply: In this new version, we have spelled out the term before the acronym.

Line 76 is repeat of line 64 with a variation: "H. capsulatum, in vitro biofilm-forming ability has been reported for filamentous and yeast forms [13,14]  "The in vitro biofilm-forming ability of this fungal species has been reported for the yeast form [13,14],

Author’s reply: We appreciate the observation; in this version, we have removed the duplicated information.

Line 79-83: "Within the host, biofilms play a crucial role in infection development, serving as a significant virulence factor by enhancing resistance against immune response mediators and antimicrobial agents. In the environment, this growth form predominates over planktonic growth,  safeguarding fungal structures from environmental threats, including UV radiation, dehydration, extreme temperatures, and chemical agents [21,22]" is again a repetition of same content from several line above it.

Author’s reply: We fully appreciate your input. We have reviewed the passage and removed the redundant information, as per your suggestion.

Line 134: Please explained how these 20 isolates were selected?

Author’s reply: We utilized 20 strains of H. capsulatum previously identified through molecular biology techniques. These strains were selected based on their prior isolation from histoplasmosis patients and their strong biofilm-forming capabilities. All strains were sourced from the fungal collection at the Center for Medical Mycology (CEMM).

Line 151: Please explain with CLSI 2020 is used? and what do you mean by "with adaptation"?

Author’s reply: We utilized the M60-2nd ed (2020) document, which standardizes susceptibility testing for yeasts. Adaptations were necessary, as the document does not address dimorphic fungal yeast, such as those of H. capsulatum. Therefore, we referred to the CLSI document and adjusted the growth times to align with our experimental conditions. These growth times, distinct from those of Candida strains, are appropriately detailed in the methodology section.

Line 237: "CLSI" Which CLSI?

Author’s reply: The M60-2nd ed (2020) document

Line 159, 165, 192, 204, 242, "All tests were conducted in duplicate at two 242 different time points". Did you come up with exact same results? I could not find any statement or statistics of these duplicate in any section of the results. Please clarify. 

Author’s reply: The results presented in the manuscript represent an average of the tests conducted in duplicate; thus, the outcome already includes the results from the duplicates.

Table 1, 2, 3 and 4: "LMW" abbreviation is used for "LMW Chitosan" and it is not self explanatory and confusing to the reader. Please come up with another abbreviation to be more clear for "Chitosan" not the LMV form of it. 

Author’s reply: We use LMW chitosan to refer to low molecular weight chitosan. We understand that this may cause confusion, and in the new version, we will use only CHI (chitosan).

Line 277, "In these combinations, the MICs of the antifungals were 4 to 16 times lower than those observed individually." This is confusing are you comparing it with the AMB and ITC alone or with LMV alone? It seems that AMB and ITC have very low MICs and in fact the combination barely improves the MIC compare to AMB or ITC alone. 

Author’s reply:  We observed the reductions for each drug separately (AMB, ITC, and LWC) and then assessed the decreases in MICs. In fact, the reductions in the antifungals AMB and ITC were more modest because their individual values were already lower. We took care to rephrase the passage to make this clear.

Table 3, line CEMM 03-0-34 under LMV/ITC it says "0.0S" Please clarify

Author’s reply:  In this case, the correct value is 0.03. In this updated version, the table reflects the correct value.

Line 290: "LWC combined with AMB or ITC reduced the MIC by up to 32 times, demonstrating the combined effects." Same as it mentioned it seems that the AMB and ITC alone have very low MIC and LMV synergistic impact is very little. So comparison is perhaps was done with the LMV alone not with AMB and ITC alone. 

Author’s reply:  You are correct; we have rephrased the passage to avoid any confusion.

Line 300: ". Synergistic interaction between LWC and AMB or ITC was observed, resulting in up to a 16-fold reduction in antifungal MIC in the presence of LWC" Same issue as above. 

Author’s reply:  We have rephrased the passage.

Line 397: "reduction in the MIC of the antifungal agent reached up to 16 times" Same issue as above.

Author’s reply: We have rephrased the passage.

Comments on the Quality of English Language

I suggest an overall English editing could improve the quality of the paper. 

Author’s reply:  As requested, we have submitted our manuscript for English language editing. This new version has been thoroughly reviewed to enhance the quality of writing in the language.

Reviewer 3 Report

Comments and Suggestions for Authors

The manuscript addresses a relevant subject and deserves publication in a journal like this one. Some concerns need to be solved though:

- No mention of how the isolates were classified as H. capsulatum. Was it by molecular techniques?

- There is no evidence that biofilms were formed following the included protocol. It seems that the biofilms are only adherent cells. Evidence of biofilm formation, such as EPS detection should be included in the study. The images in Figure 3 support these comments, there is no evidence of EPS. In the same figure, higher magnifications should be included.

- What is the degradation degree of chitosan? Some polymer molecules are likely degraded during the process or storage. So, what is the effect of glucosamine on H. capsulatum? Does it have any antifungal effect? what about chitobiose? I think these are control experiments that should be included in the study.

Comments on the Quality of English Language

The manuscript is readable but there are improvement areas in the English usage.

Author Response

Comments and Suggestions for Authors

The manuscript addresses a relevant subject and deserves publication in a journal like this one. Some concerns need to be solved though:

- No mention of how the isolates were classified as H. capsulatum. Was it by molecular techniques?

Author’s reply: Thank you very much for your considerations. Our strains were previously identified through molecular biology, and all of them are part of the mycological collection at the specialized center for medical mycology. In this new version of the manuscript, we have included this information in the methodology section.

- There is no evidence that biofilms were formed following the included protocol. It seems that the biofilms are only adherent cells. Evidence of biofilm formation, such as EPS detection should be included in the study. The images in Figure 3 support these comments, there is no evidence of EPS. In the same figure, higher magnifications should be included.

Author’s reply:  In this new version of the manuscript, we have added new images captured through microscopy, allowing the observation of the presence of EPS (extracellular polymeric substances).

- What is the degradation degree of chitosan? Some polymer molecules are likely degraded during the process or storage. So, what is the effect of glucosamine on H. capsulatum? Does it have any antifungal effect? what about chitobiose? I think these are control experiments that should be included in the study.

Author’s reply:  Thank you for the suggestions. To define the scope of the study, we conducted an extensive literature review. Through this review, we observed that various chitosan had been previously tested, including glucosamine and oligosaccharides. The authors demonstrated that glucosamine and oligosaccharides did not exhibit activity against different Candida species [1]. We believe that, based on its previous testing against other fungal pathogens, small oligomers such as glucosamine also lack antifungal activity against H. capsulatum. However, we do not dismiss the possibility of investigating potential antifungal activities of glucosamine against H. capsulatum in future studies.

In another study that served as our foundation, it was observed that chitosan molecules are highly stable, maintaining their stability even at temperatures exceeding 300 degrees Celsius. This study further indicates that compounds encapsulated in chitosan nanoparticles exhibit increased stability compared to the same compounds without chitosan [2]. We believe that the substantial stability of chitosan provides reliability in the obtained results. Additionally, in our searches, we found no evidence of enzymatic degradation performed by H. capsulatum, which may be the focus of our future studies.

Comments on the Quality of English Language

The manuscript is readable but there are improvement areas in the English usage.

Author’s reply:  As requested, we have submitted our manuscript for English language editing. This new version has been thoroughly reviewed to enhance the quality of writing in the language.

References

  1. Seyfarth, F.; Schliemann, S.; Elsner, P.; Hipler, U.-C. Antifungal Effect of High-and Low-Molecular-Weight Chitosan Hydrochloride, Carboxymethyl Chitosan, Chitosan Oligosaccharide and N-Acetyl-D-Glucosamine against Candida Albicans, Candida Krusei and Candida Glabrata. Int. J. Pharm. 2008, 353, 139–148.
  2. Garcia, L.G.S.; da Rocha, M.G.; Lima, L.R.; Cunha, A.P.; de Oliveira, J.S.; de Andrade, A.R.C.; Ricardo, N.M.P.S.; Pereira-Neto, W.A.; Sidrim, J.J.C.; Rocha, M.F.G. Essential Oils Encapsulated in Chitosan Microparticles against Candida Albicans Biofilms. Int. J. Biol. Macromol. 2021, 166, 621–632.

Round 2

Reviewer 3 Report

Comments and Suggestions for Authors

The authors addressed my previous concerns. The new Figure 3 is not very informative regarding the presence of EPS. I don't understand what are the bright focuses throughout panel A. I suggest including better figure instead.

Author Response

Dear Mr. William Zhang,

The second version of the manuscript jof-2716780  “Antifungal activity of chitosan against Histoplasma capsulatum in planktonic and biofilm forms: A therapeutic strategy in the future?”, written by Raimunda Sâmia Nogueira Brilhante1*, Anderson da Cunha Costa, Waldemiro de Aquino Pereira Neto, Jacó Ricarte Lima de Mesquita, Marcos Fábio Gadelha Rocha, Rosemeyre Freire, Augusto Feynman Dias Nobre, Mirele Rodrigues Fernandes, Gessica dos Santos Araújo, Thomas Crouzier, Ulrike Schimpf, Rodrigo Silveira Viera, follows attached to this mail. This new version was carefully revised by all authors, in accordance with reviewers’ comments. We greatly appreciate your consideration and look forward to hearing from you.

Yours sincerely,

Prof. Raimunda Sâmia Nogueira Brilhante

Reviewers’ comments and authors’ responses

Reviewers’ comments are in italics and the authors’ responses are in regular Roman style.

Reviewers' Comments to Author:

Reviewer 1:

Comments and Suggestions for Authors

The article is well written and designed. The biofilm is a current topic and manuscripts on this topic are welcome. However, I would have few suggestions in order to improve the quality of this article: 1. The Introduction section is too long, I would suggest to remove from this section the general information regarding the Histoplasma capsulatum. 2. In the Material and method section, please specify the producer of the Candida ATCC strains. 3. MIC should be expressed/calculated as µg/ml throughout the manucript.

Author’s reply: Thank you very much for your considerations. As requested, we have revised the introduction to eliminate the excess of general information about Histoplasma capsulatum. In this new version, we are also including the producer of the ATCC strains used in our studies. We have also reviewed, and in the current version, the MIC is properly expressed in µg/ml throughout the manuscript. We believe that your contribution has significantly enhanced our work, and we appreciate the time devoted to correcting our manuscript.

Reviewer 2:

These are my suggestions for corrections or requests for clarifications:

Line 31: “classical antifungal drugs showed a synergic effect, reducing chitosan’s MICs by 32 times” This is confusing “by 32 times” or is it better to mention the log reduction? Was this for the yeast-like form or hyphae?

Author’s reply: We fully agree with your suggestion. To clarify the passage in this new version of the manuscript, we chose to present the result by providing its logarithmic reduction. Additionally, we have indicated that this reduction was observed in tests involving the fungus in its yeast form.

Line 32: “demonstrating that these interactions did not exhibit any strains with antagonism” I suggest rewording.

Author’s reply: As requested, we have rephrased the passage, as can be observed in the updated version of the manuscript.

Line 54-55: “high prevalence of the infection in that region” Which region?

Author’s reply: Here we are referring to the United States.

Line 55-56: “At the moment histoplasmosis is one of the most prevalent systemic mycoses in the America” Did you mean was one of the most prevalent?

Author’s reply: The most common endemic systemic mycoses are caused by thermally dimorphic fungi such as H. capsulatum, which currently ranks among the pathogens most frequently associated with systemic diseases. In the Americas, histoplasmosis remains among the group of systemic diseases with the highest prevalence to this day. In this revised version of the manuscript, we have clarified this passage.

Line 56-61: “In recent decades, systemic and invasive fungal infections have become more common, especially among immunocompromised patients. Annually, there are an estimated 300 million cases of severe fungal infections and 1.6 million deaths worldwide [9,10]. This situation primarily arises from the rise in the count of immunocompromised patients, such as those with AIDS, cancer and transplant recipients, in addition to the increase in drug resistance, and the emergence and re-emergence of pathogens like Histoplasma capsulatum [10,11]. This section needs to be brought to the beginning of the introduction.

Author’s reply: Thank you for the suggestion; we agree and have relocated the paragraph to the beginning of the section.

Line 71: "These cells" Which cells?

Author’s reply: Here we are referring to biofilm cells, we make it clearer in the new version.

Line 76: "UV" please spell it out

Author’s reply: In this new version, we have spelled out the term before the acronym.

Line 76 is repeat of line 64 with a variation: "H. capsulatum, in vitro biofilm-forming ability has been reported for filamentous and yeast forms [13,14]  "The in vitro biofilm-forming ability of this fungal species has been reported for the yeast form [13,14],

Author’s reply: We appreciate the observation; in this version, we have removed the duplicated information.

Line 79-83: "Within the host, biofilms play a crucial role in infection development, serving as a significant virulence factor by enhancing resistance against immune response mediators and antimicrobial agents. In the environment, this growth form predominates over planktonic growth,  safeguarding fungal structures from environmental threats, including UV radiation, dehydration, extreme temperatures, and chemical agents [21,22]" is again a repetition of same content from several line above it.

Author’s reply: We fully appreciate your input. We have reviewed the passage and removed the redundant information, as per your suggestion.

Line 134: Please explained how these 20 isolates were selected?

Author’s reply: We utilized 20 strains of H. capsulatum previously identified through molecular biology techniques. These strains were selected based on their prior isolation from histoplasmosis patients and their strong biofilm-forming capabilities. All strains were sourced from the fungal collection at the Center for Medical Mycology (CEMM).

Line 151: Please explain with CLSI 2020 is used? and what do you mean by "with adaptation"?

Author’s reply: We utilized the M60-2nd ed (2020) document, which standardizes susceptibility testing for yeasts. Adaptations were necessary, as the document does not address dimorphic fungal yeast, such as those of H. capsulatum. Therefore, we referred to the CLSI document and adjusted the growth times to align with our experimental conditions. These growth times, distinct from those of Candida strains, are appropriately detailed in the methodology section.

Line 237: "CLSI" Which CLSI?

Author’s reply: The M60-2nd ed (2020) document

Line 159, 165, 192, 204, 242, "All tests were conducted in duplicate at two 242 different time points". Did you come up with exact same results? I could not find any statement or statistics of these duplicate in any section of the results. Please clarify. 

Author’s reply: The results presented in the manuscript represent an average of the tests conducted in duplicate; thus, the outcome already includes the results from the duplicates.

Table 1, 2, 3 and 4: "LMW" abbreviation is used for "LMW Chitosan" and it is not self explanatory and confusing to the reader. Please come up with another abbreviation to be more clear for "Chitosan" not the LMV form of it. 

Author’s reply: We use LMW chitosan to refer to low molecular weight chitosan. We understand that this may cause confusion, and in the new version, we will use only CHI (chitosan).

Line 277, "In these combinations, the MICs of the antifungals were 4 to 16 times lower than those observed individually." This is confusing are you comparing it with the AMB and ITC alone or with LMV alone? It seems that AMB and ITC have very low MICs and in fact the combination barely improves the MIC compare to AMB or ITC alone. 

Author’s reply: We observed the reductions for each drug separately (AMB, ITC, and LWC) and then assessed the decreases in MICs. In fact, the reductions in the antifungals AMB and ITC were more modest because their individual values were already lower. We took care to rephrase the passage to make this clear.

Table 3, line CEMM 03-0-34 under LMV/ITC it says "0.0S" Please clarify

Author’s reply: In this case, the correct value is 0.03. In this updated version, the table reflects the correct value.

Line 290: "LWC combined with AMB or ITC reduced the MIC by up to 32 times, demonstrating the combined effects." Same as it mentioned it seems that the AMB and ITC alone have very low MIC and LMV synergistic impact is very little. So comparison is perhaps was done with the LMV alone not with AMB and ITC alone. 

Author’s reply: You are correct; we have rephrased the passage to avoid any confusion.

Line 300: ". Synergistic interaction between LWC and AMB or ITC was observed, resulting in up to a 16-fold reduction in antifungal MIC in the presence of LWC" Same issue as above. 

Author’s reply: We have rephrased the passage.

Line 397: "reduction in the MIC of the antifungal agent reached up to 16 times" Same issue as above.

Author’s reply: We have rephrased the passage.

Comments on the Quality of English Language

I suggest an overall English editing could improve the quality of the paper. 

Author’s reply: As requested, we have submitted our manuscript for English language editing. This new version has been thoroughly reviewed to enhance the quality of writing in the language.

Comments and Suggestions for Authors

The manuscript addresses a relevant subject and deserves publication in a journal like this one. Some concerns need to be solved though:

- No mention of how the isolates were classified as H. capsulatum. Was it by molecular techniques?

Author’s reply: Thank you very much for your considerations. Our strains were previously identified through molecular biology, and all of them are part of the mycological collection at the specialized center for medical mycology. In this new version of the manuscript, we have included this information in the methodology section.

- There is no evidence that biofilms were formed following the included protocol. It seems that the biofilms are only adherent cells. Evidence of biofilm formation, such as EPS detection should be included in the study. The images in Figure 3 support these comments, there is no evidence of EPS. In the same figure, higher magnifications should be included.

Author’s reply: In this new version of the manuscript, we have added new images captured through microscopy, allowing the observation of the presence of EPS (extracellular polymeric substances).

- What is the degradation degree of chitosan? Some polymer molecules are likely degraded during the process or storage. So, what is the effect of glucosamine on H. capsulatum? Does it have any antifungal effect? what about chitobiose? I think these are control experiments that should be included in the study.

Author’s reply: Thank you for the suggestions. To define the scope of the study, we conducted an extensive literature review. Through this review, we observed that various chitosan had been previously tested, including glucosamine and oligosaccharides. The authors demonstrated that glucosamine and oligosaccharides did not exhibit activity against different Candida species [1]. We believe that, based on its previous testing against other fungal pathogens, small oligomers such as glucosamine also lack antifungal activity against H. capsulatum. However, we do not dismiss the possibility of investigating potential antifungal activities of glucosamine against H. capsulatum in future studies.

In another study that served as our foundation, it was observed that chitosan molecules are highly stable, maintaining their stability even at temperatures exceeding 300 degrees Celsius. This study further indicates that compounds encapsulated in chitosan nanoparticles exhibit increased stability compared to the same compounds without chitosan [2]. We believe that the substantial stability of chitosan provides reliability in the obtained results. Additionally, in our searches, we found no evidence of enzymatic degradation performed by H. capsulatum, which may be the focus of our future studies.

Comments on the Quality of English Language

The manuscript is readable but there are improvement areas in the English usage.

Author’s reply: As requested, we have submitted our manuscript for English language editing. This new version has been thoroughly reviewed to enhance the quality of writing in the language.

References

  1. Seyfarth, F.; Schliemann, S.; Elsner, P.; Hipler, U.-C. Antifungal Effect of High-and Low-Molecular-Weight Chitosan Hydrochloride, Carboxymethyl Chitosan, Chitosan Oligosaccharide and N-Acetyl-D-Glucosamine against Candida Albicans, Candida Krusei and Candida Glabrata. Int. J. Pharm. 2008, 353, 139–148.
  2. Garcia, L.G.S.; da Rocha, M.G.; Lima, L.R.; Cunha, A.P.; de Oliveira, J.S.; de Andrade, A.R.C.; Ricardo, N.M.P.S.; Pereira-Neto, W.A.; Sidrim, J.J.C.; Rocha, M.F.G. Essential Oils Encapsulated in Chitosan Microparticles against Candida Albicans Biofilms. Int. J. Biol. Macromol. 2021, 166, 621–632.
